# An apportionment method for the Oxidative Potential to the atmospheric PM sources: application to a one-year study in Chamonix, France.

Weber Samuël[1], Uzu Gaëlle[1], Calas Aude[1], Chevrier Florie[1,2], Besombes Jean-Luc[2], Charron Aurélie[1,3], Salameh Dalia[1], Ježek Irena[4], Močnik Griša[4,5], and Jaffrezo Jean-Luc[1]

[1]Univ. Grenoble Alpes, CNRS, IRD, IGE (UMR 5001), F-38000 Grenoble, France.
[2]Univ. Savoie Mont-Blanc, LCME, F-73 000 Chambéry, France.
[3]IFFSTAAR, F-69675 Bron, France.
[4]Aerosol d.o.o., Kamniška 41, 1000 Ljubljana, Slovenia.
[5]Jožef Stefan Institute, Jamova 39, SI-1000 Ljubljana, Slovenia.

*Correspondence to:* Gaëlle Uzu (gaelle.uzu@ird.fr)

**Abstract.** Inhaled aerosolized particulate matter (PM) induces cellular oxidative stress in vivo, leading to adverse health outcomes. The oxidative potential (OP) of PM appears to be a more relevant proxy of the health impact of the aerosol rather than the total mass concentration. However, the relative contributions of the aerosol sources to the OP are still poorly known. In order to better quantify the impact of different PM sources, we sampled aerosols in a French city for one year (year 2014, 115 samples). A coupled analysis with detailed chemical speciation (more than 100 species, including organic and carbonaceous compounds, ions, metals and Aethalometer measurements) and two OP assays (ascorbic acid (AA) and dithiothreitol (DTT)) in a simulated lung fluid (SLF) were performed in these samples. We present in this study a statistical framework using a coupled approach with Positive Matrix Factorization (PMF) and multiple linear regression to attribute a redox-activity to PM sources. Our results highlight the importance of the biomass burning and vehicular sources to explain the observed OP for both assays. In general, we see a different contribution of the sources when considering the OP AA, OP DTT or the mass of the $PM_{10}$. Moreover, significant differences are observed between the DTT and AA tests which emphasized chemical specificities of the two tests and the need of a standardized approach for the future studies on epidemiology or toxicology of the PM.

## 1 Introduction

Exposure of the population to pollution by airborne particles is a growing concern due to its burden on human health, ranking as the 5th risk factor for total deaths from all causes across ages and sexes in 2015 (Cohen et al., 2017). Such impact is assessed through cross-over studies based on health data and particulate matter (PM) mass concentrations (Pope III, 2004; Pope III and Dockery, 1999; WHO, 2016). However, the dominant fraction of the PM mass are ionic species or crustal elements and these contribute little to PM toxicity (Ayres et al., 2008). Therefore, new metrics are currently investigated in order to better quantify the effect of the population exposure. Among the different metrics, oxidative potential (OP) addresses the intrinsic capacity of PM to generate Reactive Oxygen Species (ROS) able to oxidize the lungs. It has been proposed as a unifying factor for

quantifying the effects of particulate exposure as it relies on surface area, size and PM composition (Ayres et al., 2008; Sauvain et al., 2009; Kelly and Fussell, 2012; Gehling and Dellinger, 2013; Sauvain et al., 2013; Fang et al., 2016; Crobeddu et al., 2017; Abrams et al., 2017).

Many methodologies to quantify OP exist, and none has become a standard so far. Since each OP methodology is somewhat specific to the precise type of ROS or ROS-inducer (Yang et al., 2014), a standard methodology should most probably include several assays in order to fully determine the ROS generation propensity (Janssen et al., 2015; Sauvain et al., 2013). Such a combination has not emerged yet, since the link between OP and chemical composition of PM is not fully understood, and OP drivers are not truly supported by evidence.

Investigating the link between OP and chemistry of PM is not simple, since particles chemical composition is unique in every sampling point. Moreover, univariate correlations can lead to false resultsr. For example, strong OP correlation with polycyclic aromatic hydrocarbon (PAH) can be found within dithiothreitol (DTT) assay (Calas et al., 2018). This correlation is chemically impossible since DTT, a reducing agent, needs redox-active compounds to be depleted (Ntziachristos et al., 2007; Shirmohammadi et al., 2016). This correlation is now well explained since PAHs are co-emitted with quinones, oxy-PAH which are redox-active and able to oxidize DTT (Charrier et al., 2015; Charrier and Anastasio, 2012). Linear multiple regression is not trivial to use in determining OP factors, since extreme outliers need to be removed, normal distributions are needed and negative contributions may be attributed to mathematically explain annual OP variations (Calas et al., 2018).

Another option is to consider the sources contribution instead of the chemical species (Verma et al., 2014; Bates et al., 2015; Fang et al., 2015, 2016). Indeed, working directly with chemical species involves assessing an exhaustive composition characterization. This is rather impossible since many species in the complex mixture of aerosols remain unidentified. Moreover, if a detailed composition (which can sometimes include up to 150 species (Waked et al., 2014)) is provided, at least the same number of samples for OP measurements is needed, otherwise, the system remains underdetermined. Reducing the system by direct truncation is not possible since species contributing to OP could be dropped, inducing some degree of unknown bias. Conversely, if the explanatory variables are the sources' contributions, biases are mitigated. However, the sources dynamics need to be determined for a long period of time in order to reflect the climatology of the location. Moreover, the composition of a given named source may vary according to its location (Belis et al., 2013). To mitigate theses issues, we decide to use a PMF approach instead of a CMB model to better render the local specificities of the sources. Indeed, the CMB averages the sources profile's from different studies and is then locally biased. Furthermore, in this study a whole year of analysis is used as input of the PMF. We then have a climatological view of the sources dynamics.

The objective of this study is to present a methodology for the evaluation of the contributions of common sources of particles to the overall OP for a long time series of $PM_{10}$ sample (PM with a diameter lower than 10 µm). The OP was measured on filter samples collected over a full year in the city of Chamonix (Alpine valley), using two OP protocols: the ascorbic acid (AA) and dithiothreitol (DTT) assays. An inversion procedure of these OP measurements was developed using source apportionment results obtained from an advanced source-receptor model PMF (Chevrier, 2016), in order to attribute both an intrinsic OP to the sources and the evolution of the sources contributions to OP's over the year.

## 2 Methods

This work takes advantage of an already existing database, based on Particulate Matter ($PM_{10}$) samples collected during the DECOMBIO program (Chevrier et al., 2016), with the chemical analyses and the source apportionment of PM having already been conducted (Chevrier, 2016), and the OP measurements performed on the same samples (Calas et al., 2018). These are briefly presented below.

### 2.1 Site and sampling

Sampling took place in the city of Chamonix-Mont-Blanc ($45°55.358'$ N, $6°52.194'$ E), in the Alpine Arve Valley, below Mont-Blanc (Figure 1). The sampling site is located in the middle of the town, in a densely populated area, with the sampling cabin being close to a street. A one-year study was conducted from November, $14^{th}$ 2013 to October, $31^{th}$ 2014, with 24 hour $PM_{10}$ sampling taking place every third day, giving a series of 115 samples. These daily $PM_{10}$ samples were collected with a high volume sampler (Digitel, DA80, $30 \, m^3 \, h^{-1}$) on pre-fired quartz filters (Pall, Tissuquartz). All details concerning the site and the logistical aspects of the sampling procedure can be found in Chevrier (2016).

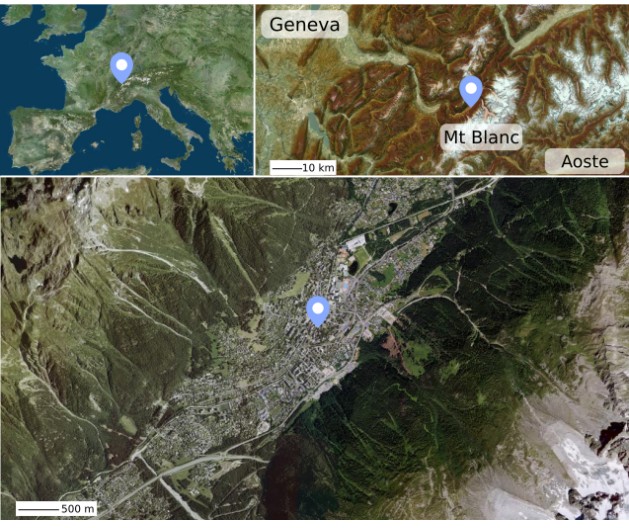

**Figure 1.** Location of the sampling site in Chamonix, in the Arve Valley, France ($45°55.358'$ N, $6°52.194'$ E). © Planete observer, IGN

### 2.2 Chemical analyses

All filters were analyzed using a large array of methods for the quantification of chemical species including those important for the mass balance of the PM (EC, OC, ions...) and many organic and inorganic tracers of sources. The elements and components analyzed are:

- Organic and elemental carbon (OC, EC), using a Sunset instrument and the EUSAAR2 protocol (Aymoz et al., 2004; Cavalli et al., 2016);

- soluble anions and cations ($NO_3^-$, $SO_4^{2-}$, $Cl^-$ and $NH_4^+$, $Mg^{2+}$, $Na^+$, $Ca^{2+}$, $K^+$) through ionic chromatography (Waked et al., 2014);

- inorganic elements (Al, Fe, Ti, As, Ba Cd, Ce, Cr, Cu, La, Li, Mn, Mo, Ni, Pb, Rb, Sb, Sn, Sr, V, Zn and Zr) using ICP-MS (Waked et al., 2014);

  - sugar alcohols (arabitol, sorbitol, and mannitol, also called polyols) and anhydrous monosaccharides (levoglucosan, mannosan and galactosan) using an HPLC-PAD method (Waked et al., 2014);

  - polar and nonpolar organic tracers (alkanes, hopanes, methoxyphenols, and substituted derivatives (methyl-PAHs) and
  polycyclic aromatic sulfur heterocycles (PASHs) using GC-MS and polycyclic aromatic hydrocarbons (PAHs) using HPLC-fluorescence (Golly et al., 2015).

Additionally, Black Carbon (BC) measurements were ongoing throughout the year, with wood burning $BC_{bb}$ and fossil fuel $BC_{ff}$ fractions determined using an Athalometer AE33 and the so called "Aethalometer model" (Sandradewi et al., 2008; Drinovec et al., 2015). Although the BC measurements was performed on $PM_{2.5}$ samples, we decided to use it as Jaffrezo et al.
(2005); Cavalli et al. (2016) show that the amount of EC in $PM_{10}$ and $PM_{2.5}$ is almost equivalent.

All the procedures for these chemical analyses are described in detail in Chevrier (2016).

## 2.3   Source apportionment of $PM_{10}$

The source apportionment was performed with Positive Matrix Factorization, using the US EPA software PMF 5.0 (US EPA, 2017), following the recommendations included in the european guideline book issued in the EU Fairmode program (Belis
et al., 2014). However, in the environment of Alpine valleys, the local meteorology and frequent inversion layers in winter lead to strong covariations of the concentrations of many chemical species emitted from the valley bottom. Indeed, during temperature inversion in Alpine valley, pollutants are stuck into the Atmospheric Boundary Layer (ABL) and cannot be removed by wind. Such inversion may be stable during several days. As a result, the different emission sources during that period of time add together and the dynamic from the different sources is masked. In other words, one sample does not integrate anymore
only emissions during the sampling time, but also emissions of the previous days. This end-up with chemical species in one sample that should not be present together, respect to the temporality of their respective sources. Thereby, their correlation is increased. The covariation of the different pollutants adversely influences the ability of PMF to distinguish different sources. Therefore, we developed an approach including several specificities, rarely applied in classic source apportionment, in order to overcome this methodological problem in the PMF (Chevrier, 2016).
First, many tracers were included as input parameters, including specific organic tracers. The benefit of such approach was previously described (Golly, 2014; Waked et al., 2014; Srivastava et al., 2017). In our case, we included hopanes (thereafter named HOP), methoxyphenols, polyols (sum of mannitol, arabitol and sorbitol), levoglucosan and MSA (methane sulfonic

**Table 1.** Selection of the chemical species used as input variables in the EPA PMF5.0 model and their relative uncertainties. Σpolyols refers to the sum of arabitol, sorbitol and mannitol and Σmethoxyphenol to the sum of the particulate methoxyphenols. The uncertainties in "%" are relative to the sample concentration for the species.

| | Total | Carbonaceous matter | | Ions | Organics compound | | Metals |
|---|---|---|---|---|---|---|---|
| Species | $PM_{10}$ | OC* | $BC_{bb}$, $BC_{ff}$ | $Cl^-$, $NO_3^-$, $SO_4^{2-}$, $Na^+$, $NH_4^+$, $K^+$, $Mg^{2+}$, $Ca^{2+}$ | Σpolyols, MSA, levoglucosan | ΣHOP, ΣMethoxyphenol | As, Cu, Fe, Mn, Mo, Ni, Pb, Rb, Sb, Ti, V, Zn, Zr |
| Uncertainty | 20% | 10% | 20% | Gianini et al. (2013) | 15% | Gianini et al. (2013) | 2×Gianini et al. (2013) |

acid). Instead of OC we used the difference (OC*) between the OC and the carbon equivalent of these previously analyzed species.

Second, elemental carbon (EC), which is an important species for the deconvolution of combustion sources was in the PMF replaced by $BC_{bb}$ and $BC_{ff}$ obtained using the "Aethalometer model" by concurrent measurements with the Aethalometer AE33. This provides a very strong information on the sources, as already pointed out in other studies (Petit et al., 2015). No correction was introduced to compensate between EC and BC (Zanatta et al., 2016).

Finally, we took advantage of the possibilities of PMF 5.0 to apply constraints to the factor profiles, in order to better define the sources (Golly, 2014; Srivastava et al., 2017; Salameh, 2015). A minimal set of constraints based on prior and external geochemical knowledge of sources fingerprints was applied:

- in the biomass burning factor, the contributions of levoglucosan, potassium, methoxyphenols and $BC_{bb}$ were increased, whereas the $BC_{ff}$ and HOP were set to 0,

- HOP was increased in the vehicular factor.

We increased the concentration of the species in the factors thanks to the "pull up maximally" option of the EPA PMF5.0 software (US EPA, 2017), which tried to increase the contribution of the given specie to the factor. Table 1 sums up the input chemistry species and respective uncertainties used in the PMF study.

## 2.4 Measurements of the Oxidative Potential of PM

The methodology is described in detail in Calas et al. (2017). In brief, we performed the extraction of PM into the simulated lung fluid (SLF) solution to simulate the bio-accessibility of PM and to closer simulate exposure conditions. The extraction took place into SLF at iso-mass. All samples were analyzed at $10\,\mu g\,mL^{-1}$ of PM, by adjusting the area of filter extracted. The filter extraction method includes both water soluble and insoluble species. After the SLF extraction, particles removed from filter are not filtrated; the whole extract is injected in the multiwall plate. Samples were processed using the AA and DTT assays. DTT depletion when in contact with PM extracts was determined by dosing the remaining amount of DTT with DTNB (dithionitrobenzoic acid) at different reaction times and absorbance was measured at 412 nm using a plate spectrophotometer

(Tecan, M200). The AA assay is a simplified version of the synthetic respiratory tract lining fluid (RTFL) assay (Kelly and Mudway, 2003), where only AA is used. AA depletion is read continuously for 30 min from absorbance at 265 nm (TECAN, 200). The maximum depletion rate of AA is determined by linear regression of the linear section data. For both assays, the 96-wells plate is auto shaken for 3 seconds before each measurement and kept at 37 °C. Three filter blanks (laboratory blanks) are included in every plate (OP AA and OP DTT) of the protocol. The average values of these blanks are then subtracted from the sample measurement of this plate. LOD value is defined as three times of the standard deviation of laboratory blanks measurements (blank filters in Gamble+DPPC solution).

The samples were stored 3 years before analyzed. As mentioned in Verma et al. (2015), the OP activity may be impacted by such storage time. However, in a previous program (ANSES ExPOSURE, 2017), still ongoing, we have been measuring the same filter over time. After one year, OP results for AA assay remain equivalents. DTT results showed a regular decrease of 15 % the first 6 months before stabilization.

Only 98 samples out of the 115 collected were measured for OP, removing samples with insufficient PM mass concentration ($<5\,\mu g\,m^{-3}$) that did not afford filter extraction at $10\,\mu g\,mL^{-1}$. The oxidative potential (OP) unit is then expressed in nmol per minute per microgram of PM. However, the population exposition is (in the first order) proportional to the mass of the inhaled PM. Therefore, the OP per microgram was multiplied by the total mass concentration of PM ($\mu g\,m^{-3}$) in order to express the OP in unit of $nmol\,min^{-1}\,m^{-3}$. We should however keep in mind that this measurement of OP may not be the exact OP from PM inhaled by the population, since we suppose a linear relationship between the OP per µg of PM and the OP of the total amount of PM. Indeed, some cocktail effects like complexation or chelation may occur for PM concentrations higher than the one tested. It has been shown by Calas et al. (2017) that the result is generally a probable over-estimation of the "true" OP. Hereafter, the OP normalized by volume is denoted with a subscribed v (OP $AA_v$ and OP $DTT_v$).

## 3 Results

### 3.1 Evolution of the OP

Both assays present a strong seasonality (Fig. 2) as already mentioned in Calas et al. (2017), and both the OP $AA_v$ and OP $DTT_v$ results show seasonality. The $OP_v$ remains high during winter and low during summer. This observation tends to emphasize the importance of PM sources which also show distinct seasonality. However, we can also observe fast variations from day to day, which may be related to a change in the PM chemistry or composition or a change in PM concentration related to sources or meteorological conditions.

Despite both assays following the same annual trend, some significant differences exist. For instance, during summer, OP $DTT_v$ shows larger values, whereas the values of OP $AA_v$ are close to 0. Moreover, the variation of the OP $AA_v$ seems smoother than that of OP $DTT_v$, especially during summer and fall (May to November). This underlines that the assays are sensitive to different chemical species present in PM.

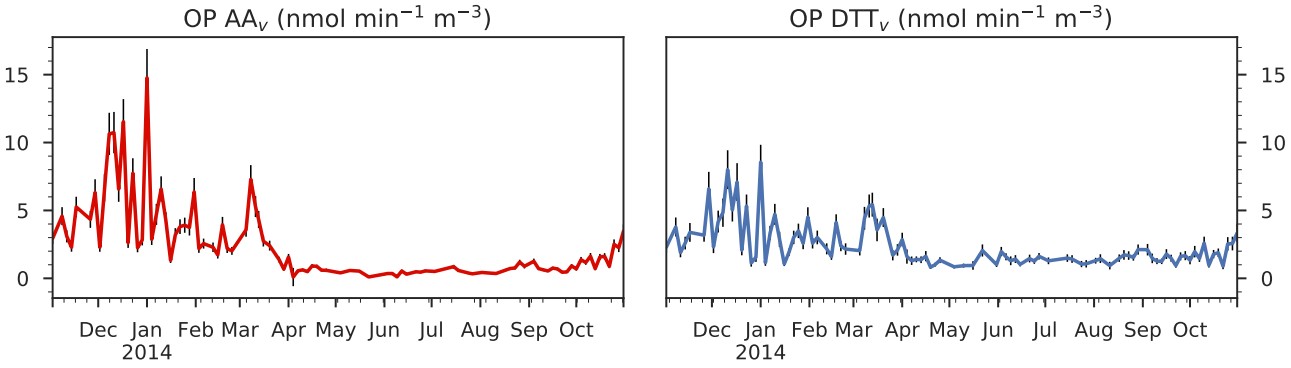

**Figure 2.** OP $AA_v$ and OP $DTT_v$ variation from 2 November 2013 to 31 October 2014 (98 samples) at the Chamonix station. The error bars represent the uncertainties (standard deviation) of the measurement. The OP unit is normalized by volume and is expressed in $\mathrm{nmol\,min^{-1}\,m^{-3}}$.

## 3.2 Evolution of the sources contributions

The PMF was already thoroughly discussed in Chevrier (2016). Briefly, 8 sources were identified: biomass burning, crustal dust, nitrate rich, sulfate rich, primary biogenic emissions an secondary biogenic aerosol, salt and vehicular emissions. Their respective main chemical species and related information are provided in the Supplementary Information (SI 1). We mainly

see that metals (notably copper) and some organics species are highly correlated to both OP, together with many fractions of the carbonaceous matter (OC, $BC_{bb}$ and $BC_{ff}$, see SI 2). Figure 3 presents sources contributions to PM. The dominant PM source is biomass burning during winter with some daily concentrations exceeding $40\ \mathrm{\mu g\,m^{-3}}$. The primary and secondary biogenic sources are mainly active during summer, as is the sulfate rich source. The vehicular source is quite constant all over the year. Indeed, the higher concentration during winter may be attributed to accumulation in the ABL, and not to an increase

of emission. The crustal dust contribution is sporadic and could include some Saharan episodes (Aymoz et al., 2004). Finally, the salt source is low but presents a high spike during March, being maybe related to road salting at that time of the year. The correlation between the OP and the sources are presented and discussed in the SI 2. Briefly, the vehicular and biomass burning sources appear to be strongly correlated to both OP ($r > 0.8$). The nitrate-rich factor presents a lower correlation, as well as the sea/road salt one ($0.3 < r < 0.6$ for both OP's), whereas the secondary biogenic, primary biogenic, and sulfate-rich factors

are slightly anti correlated with both OP's ($-0.6 < r < -0.3$). Crustal dust correlation is not significant with respect to the AA test but presents low correlation to the DTT test ($r = 0.15$ and $r = 0.35$, respectively).

## 3.3 Setting up a multiple linear regression

As the OP is a value of reactivity, it cannot be directly introduced in a mass-balance model. Hence, in order to estimate the contributions of the PM sources to the OP, we must use an inversion method. Despite the possible non linearity of OP values

with increasing masses of PM, as discussed below, we assume in this work that the OP is linearly linked to the mass. Thus,

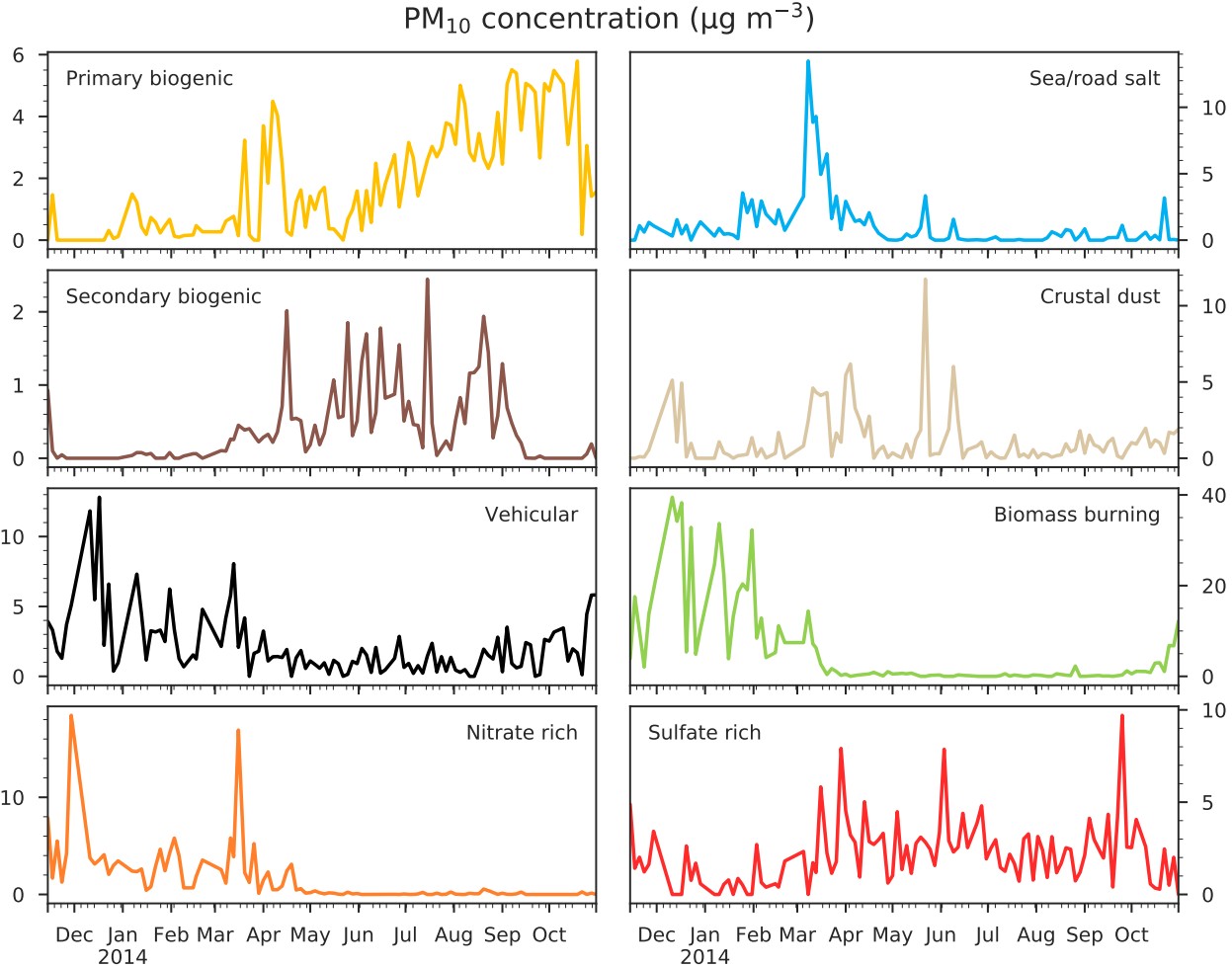

**Figure 3.** Mass concentrations of the eight PMF sources as fractions of $PM_{10}$ from 14 November 2013 to 31 October 2014 (107 samples) at the Chamonix station. Units are expressed in $\mu g\,m^{-3}$. Note different scales on the source contributions.

we hereafter assume that OP and the explanatory variables, namely the mass of the PM sources $m_{PM}$, are linearly related as follows

$$OP_{obs} = m_{PM} \cdot \beta + \varepsilon \tag{1}$$

where $OP_{obs}$ is the $(n \times 1)$ observed OP matrix in $nmol\,min^{-1}\,m^{-3}$, $m_{PM}$ the $(n \times (p+1))$ matrix with the PM mass attributed to each source expressed in $\mu g\,m^{-3}$ and a constant unity term with no unit for the intercept, and $\varepsilon$ the $(n \times 1)$ uncertainty matrix in $nmol\,min^{-1}\,m^{-3}$; n is the number of samples and p the number of sources. The estimator $\beta$ (matrix $(p+1) \times 1$) represents the intrinsic OP of the sources (i.e. the OP per mass unit of PM attributed to a given source) and the intercept, expressed respectively in $nmol\,min^{-1}\,\mu g^{-1}$ for the intrinsic OP and in $nmol\,min^{-1}\,m^{-3}$ for the intercept.

The optimal approximation of a solution for the linear system expressed by equation 1 is typically found by least squares. A variety of methods exist differing on the function to be minimized and on the regularization or sparsity penalty imposed to perform variable selection on $\beta$, from ordinary least squares to ridge regression and LASSO (least absolute shrinkage and selection operator). Here we have chosen a Weighted Least Squares (WLS) approach as it has an integrated way to handle the OP uncertainties. We have also chosen not to add a penalty function as we do not have prior knowledge on the intrinsic OP values. However, regular WLS do not rule out negative solutions, which should be implemented in our case since it is not demonstrated that intrinsic OP negative values exist in the real world. Therefore, a stepwise regression is conducted. The underlying algorithm is

1. Solve the WLS problem and estimate the intrinsic OP,

2. If an intrinsic OP is negative, then set it to zero and go back to step 1,

3. Repeat until all intrinsic OPs are positive or zero.

No source is discarded based on its p-value but only if its intrinsic OP is negative. We did not choose a direct non-negative least square approach as it would be a constraint in the model we believe is too strong. In addition, we can use the absence of negative coefficients as a test for the coherence of our dataset. Such approach may allow us to investigate which sources present a negative OP and why. This loop converges in a finite number of iterations, either to a situation with zero sources – which would be discarded as absurd, pointing to a breakdown of the underlying assumptions, or to an acceptable solution with a lower number of sources. In our particular case, since OP measurements never display negative values or negative source contributions from the PMF, the method is strictly guaranteed to converge to an acceptable solution. Further, we expressly do not set the intercept to zero in Eq. 1, choosing instead to use this as a check on our method. If the system is well constrained (i.e. no missing sources) the intercept should be close to zero within the model uncertainties, without any explicit constraint. The reciprocal situation could point to missing explanatory variables.

The uncertainties of the intrinsic OP are extracted from the variance of $\beta$, which in turn is derived from the Hessian matrix of the WLS regression in the standard way. However, the uncertainties on the modelled OP are not analytically computed. Indeed, some coefficients present co-variation due to the activity of the sources in the very same period of the year, so analytical variance cannot be used to estimate uncertainties. Therefore, in order to estimate the uncertainties of the modelled OP, we bootstrap the solution $\beta$ 1000 times with a Monte-Carlo algorithm. The bootstrap simply selects randomly an intrinsic OP for each source according to their respective normal distribution.

The algorithm was implemented in Python 3 making use of the *statsmodels* WLS module (Seabold and Perktold, 2010).

The method proposed here is an improvement of the one of Bates et al. (2015) and our methods differ in several points. First of all, our backward elimination criteria is based on the negativity of a source and not in its p-value. Indeed, a source might present a statistically significant negative value. But according to us, a source with negative intrinsic OP does not have a geo-chemical sense as the air is known to be a strong oxidant milieu. Secondly, as Bates et al. (2015) didn't measure the uncertainty of their OP samples, they used an ordinary least square (OLS) regression. On the opposite, we have an estimation

of our measurements uncertainty thanks to triplicate. We then use a weighted least square (WLS) regression instead. Finally, we propose a way to estimate uncertainties of our estimated OP with a Monte-Carlo method, which is not provided in the previous study. Moreover, the method proposed here does not only include the multiple linear regression (MLR) but also the use of the PMF model instead of the CMB one. Indeed, the MLR is highly sensitive to the explanatory variable and we decide
to use the local sources' profile (PMF) instead of the chemical mass balance method with ensemble-averaged source impact profiles.

## 3.4 Application to the Chamonix site and discussion

Figure 4 shows the comparison between observed and modelled OPs for the measurements at the Chamonix site for both OP AA and OP DTT assays. Table 2 presents the intrinsic OP AA and OP DTT in $\mathrm{nmol\,min^{-1}\,\mu g^{-1}}$ for each source with their
respective uncertainties and p-values.

### 3.4.1 Accuracy of the model

The method developed in this study appears to be sufficiently accurate to explain the two OP annual series at Chamonix. First, the seasonal trend of the OP is very well reproduced, despite some under-estimation of some of the highest values in winter. Second, the intercept of the equation regression for OP $AA_v$ is not significant (p>0.05). It is not so clear for the DTT test, but
the p-value remains high (p=0.04). We can consider the intercepts of the equation regression as nearly negligible (see Table 2). The PM sources presented in this study are then sufficient to explain the observed OP $AA_v$ and OP $DTT_v$ time series. We can also note that none of the sources was excluded for the DTT assay during the inversion procedure due to negative contributions. It emphasizes the fact that the sources explain well the observed OP. However, one source was discarded for the AA assay: the crustal dust. Its p-value was less than 0.01 for an intrinsic OP of $-0.05\pm0.01\,\mathrm{nmol\,min^{-1}\,\mu g^{-1}}$. We supposed that the crustal
dust source in this study is a mixing of several sources, including Saharan dust and road suspension dust. We could then end-up with a mixing of highly different redox-active compounds towards AA test that could explain the error for this source. Further work is needed to understand this behavior.

### 3.4.2 Uncertainties and residual

The uncertainties of the modeled OP are quite low and mostly in the range of the measurement uncertainties (Figure 4).
Indeed, the distribution of the residual is close to the normal law (Figure 5). However, we can note an asymmetry toward underestimation and residual seem to increase almost linearly with the endogenous variable (no random repartition around 0 for the highest OP). The OP is underestimated by the model for these days featuring concentrations. It may suggest either non-linearity with high loading (as suggested above), or particular event that is not apportioned by the sources provided in this study.

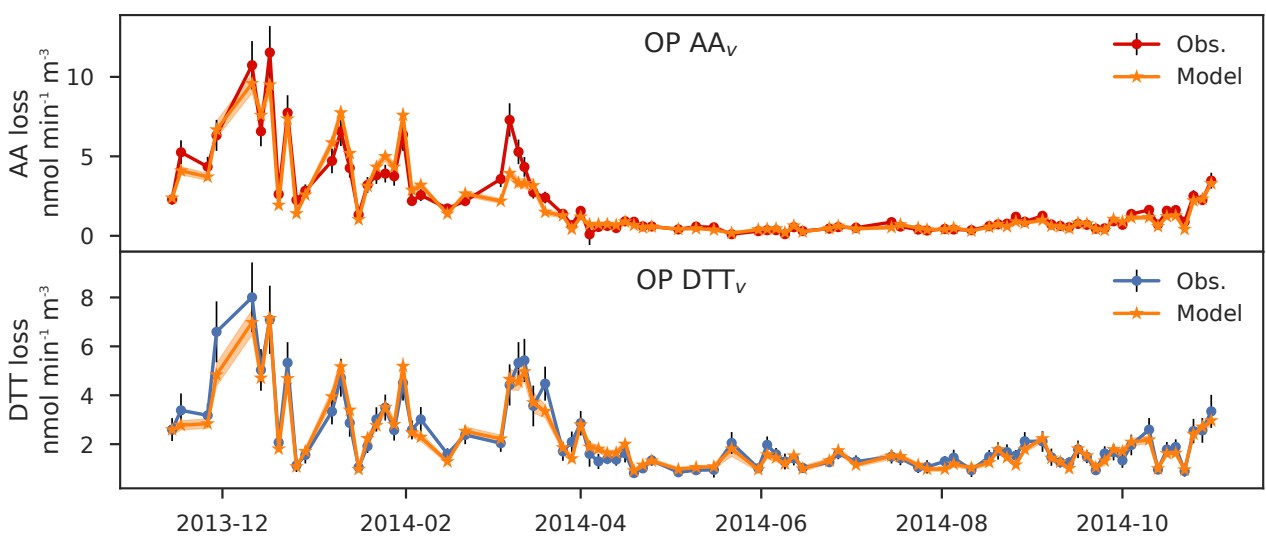

**Figure 4.** Comparison of the modelled OP (orange) and the observed OP for both the AA (top graph) and DTT (bottom graph) test (85 samples) from November 2013 to November 2014. The black error bars are the standard deviation of the observed values and the shaded orange area the uncertainties of the modelled OP. Units are in OP normalized by volume and expressed in $\mathrm{nmol\,min^{-1}\,m^{-3}}$.

### 3.4.3 Intrinsic OP

Values of the intrinsic OP of different sources for both the AA and DTT assays are ranging from zero to $0.18\pm0.01\,\mathrm{nmol\,min^{-1}\,\mu g^{-1}}$ for the AA test and from $0.06\pm0.02\,\mathrm{nmol\,min^{-1}\,\mu g^{-1}}$ to $0.27\pm0.03\,\mathrm{nmol\,min^{-1}\,\mu g^{-1}}$ for the DTT test (Table 2). The various sources do not have the same reactivity toward the AA and DTT. We also note that the two tests present different intrinsic OP

5    for the same source, and the relative importance of the sources differs from one test to the other. For instance, the vehicular source displays a lower intrinsic OP ($0.15\,\mathrm{nmol\,min^{-1}\,\mu g^{-1}}$) than the biomass burning ($0.18\,\mathrm{nmol\,min^{-1}\,\mu g^{-1}}$) for the AA test but a higher intrinsic OP for the DTT test ($0.27\,\mathrm{nmol\,min^{-1}\,\mu g^{-1}}$ for the vehicular and $0.07\,\mathrm{nmol\,min^{-1}\,\mu g^{-1}}$ for the biomass burning). This deconvolution method may be able to account for the chemical specificity of the two OP assays. In addition, the DTT test seems to be more multi-sources influenced than the AA test.

10    Nevertheless, we clearly see the importance of the vehicular source, which is associated to a strong intrinsic OP in both the AA and DTT assays. Previous studies (Bates et al., 2015; Fang et al., 2016; Verma et al., 2014) also highlighted the importance of this source to explain the OP AA and DTT. We may explain such high intrinsic OP by the presence of metals in this source – notably the copper (Charrier et al., 2015).

In the AA assay, the biomass burning also presents a high redox-activity per µg of PM. This result disagrees with Fang

15    et al. (2016) as they found no activity for this source in the OP AA test. Such difference for the biomass burning source may be explained by the two extraction protocols (in water or in a SLF solution) or by the proximity of the biomass source in Chamonix compared to the longer distance transport in Atlanta, that would change the chemistry of the source profile.

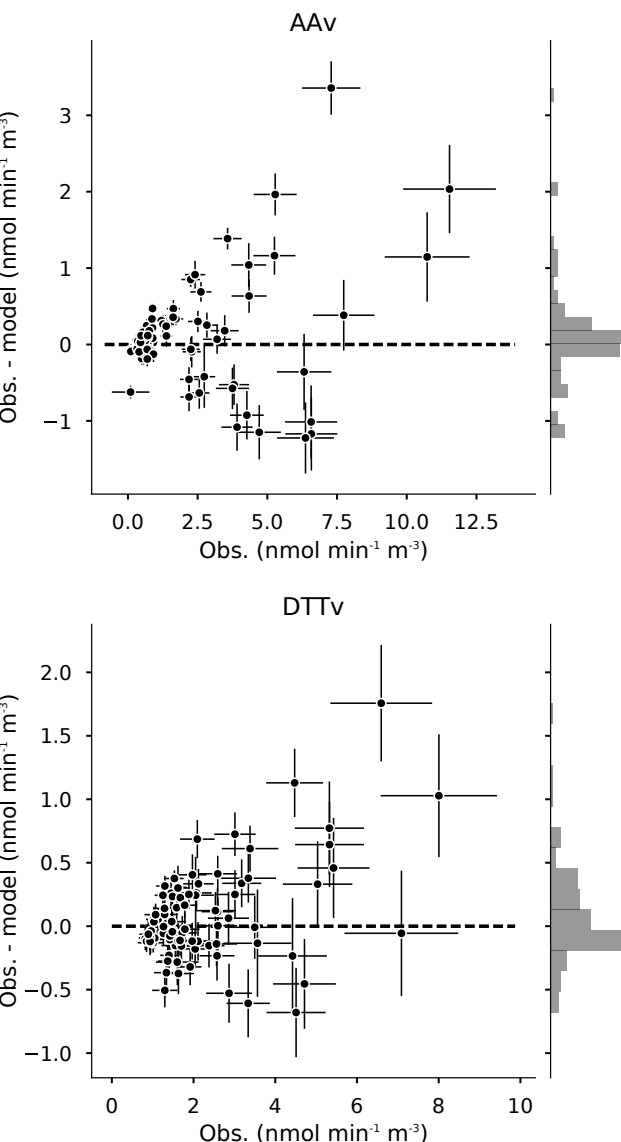

**Figure 5.** Residual distribution for the regression of the AA and DTT assays (85 samples). The error bars represent the standard deviation of the observation and the model. The histogram on the right is the distribution of the residuals. Units are in OP normalized by volume and expressed in $\mathrm{nmol\,min^{-1}\,m^{-3}}$. Note different scales for the $AA_v$ and the $DTT_v$.

However, the biomass burning in the OP DTT test has an intrinsic OP of $0.07\pm0.01$ $\mathrm{nmol\,min^{-1}\,m^{-3}}$, which is coherent with the previous study of Fang et al. (2016). The presence of oxygenated compounds such as quinones which are redox-active in the organic matter could explain this high intrinsic OP.

The nitrate-rich source also appears to contribute in the redox-activity in both assays. Although the nitrate itself is not redox-
5 active, it can be present with species that are oxidants. More work is needied in order to understand the evolution of intrinsic

**Table 2.** Regression coefficients (i.e. intrinsic OP) expressed in $\mathrm{nmol\,min^{-1}\,\mu g^{-1}}$ at Chamonix for the AA and DTT assays. The values are the mean±standard deviation based on N=1000 bootstrap of the best solution. The p-value is in the parenthesis. The crustal dust source was excluded during the inversion process for the AA test.

| | Biomass burning | Crustal dust | Nitrate rich | Primary biogenic | Sea/road salt | Secondary biogenic | Sulfate rich | Vehicular | Intercept |
|---|---|---|---|---|---|---|---|---|---|
| Unit | | | | $\mathrm{nmol\,min^{-1}\,\mu g^{-1}}$ | | | | | $\mathrm{nmol\,min^{-1}\,m^{-3}}$ |
| AA | 0.18±0.01 | - | 0.12±0.02 | 0.07±0.01 | 0.03±0.01 | 0.02±0.04 | 0.00±0.01 | 0.15±0.02 | 0.05±0.08 |
| | (<0.001) | (-) | (<0.001) | (<0.001) | (0.140) | (0.598) | (0.942) | (<0.001) | (0.502) |
| DTT | 0.07±0.01 | 0.07±0.02 | 0.07±0.02 | 0.12±0.02 | 0.14±0.03 | 0.18±0.05 | 0.06±0.02 | 0.27±0.03 | 0.17±0.08 |
| | (<0.001) | (0.003) | (<0.001) | (<0.001) | (<0.001) | (<0.001) | (<0.001) | (<0.001) | (0.045) |

OP for the nitrate rich factor, including measurements on series characterized by specific spring events related to agricultural activities, and series close to traffic sites for $NO_x$ emissions.

The primary biogenic source, mainly identified by the presence of polyols, presents a significant intrinsic OP. This result was unexpected. Indeed, Liu et al. (2010) shows that mannitol is a strong anti-oxidant. Our result suggests that some chemical species, present in the primary biogenic source but not measured in this study, may contribute to the OP of the PM from this source. Recently, it was shown that fungal spores exhibit a significant intrinsic OP (Samake et al., 2017), and this may be an hypothesis to be further tested. However, the PMF profile of the primary biogenic source may also be a mixing of different sources in our study (there is $BC_{ff}$ in it for instance). Such mixing may also explain the high intrinsic OP of this source.

Nevertheless, all these results contrast with those from simple univariate correlations between OP and sources. Indeed, the secondary biogenic source which is slightly anti-correlated to both OP's is in fact the second most redox-active source when considering intrinsic OP DTT. On the contrary, the sulfate-rich factor is slightly anti-correlated to the OP $AA_v$ but present an intrinsic OP AA close to 0. The vehicular factor, which highly correlates with OP's is also the dominant source in terms of intrinsic OP's for both assays. Such results emphasize the real interest to go replace the simple univariate correlation by a more comprehensive statistical analysis when considering the contribution of the sources (or species) to the OP's.

### 3.4.4 Contribution to the OP

The aim of this study was to establish a deconvolution model for the OP. The results obtained with it will be discussed in depth in another study, including other sites. However, here are some preliminary results for the Chamonix station concerning the sources contribution.

Due to the different intrinsic OP of the sources, the source contributions to the OP (intrinsic OP times by the source contribution in $\mathrm{\mu g\,m^{-3}}$) is different from their contribution to the PM mass. Figure 6 illustrates the normalized contribution of the sources to the mass of the aerosols and the OP measures with AA and DTT. It shows that the vehicular source barely contributes to 17 % of the total PM mass during March-April-May (MAM) and June-July-August (JJA) but more than 30 %

to the OP DTT$_v$ in the same period, and even reaching around 50 % of the OP AA$_v$ in JJA. Conversely, some sources largely contributing to the PM$_{10}$ mass such as the sulfate-rich source (30 % of the total PM mass in JJA) do not contribute to the OP (2 % to the OP AA$_v$ in the same period). Finally, some sources like the biomass burning contribute to a large extend in both PM mass and OP (on an annual basis: 35 % of the PM mass, 55 % of the OP AA$_v$ and 22 % of the OP DTT$_v$). We also note

that on an annual basis, the contribution of the vehicular source is much larger for both OP assays than for the mass. All these outcomes are key parameters for policy initiatives.

    To sum up, with this methodology, we observe a redistribution of the relative importance of the sources ranked as ROS contributors. This study gives, and more generally, the OP gives us a new vision of the atmospheric aerosols and associated ROS burden. We also point out a clear distinction between the different OP tests. Such differences raise new questions on OP

assays choices and standardization and require further investigation, especially coupled OP-toxicology-epidemiology studies.

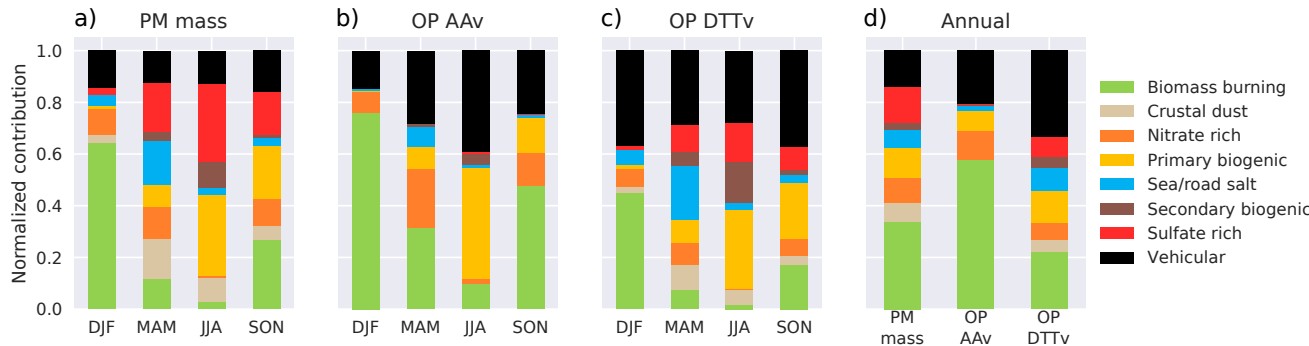

**Figure 6.** Normalized seasonal contribution of the sources to a) the PM$_{10}$ total mass, b) the OP AA$_v$ and c) the OP DTT$_v$. DJF is December-January-February, MAM is March-April-May, JJA is June-July-August and SON is September-October-November. d) annual normalized contributions of each source to the PM, OP AAv and OP DTTv.

## 4    Limitations

First of all, when comparing with others previous studies we should note that our PM extraction of samples were done in a SLF and not in water. This induces a difference in OP measurement which is not predictable for the complexes occurring between PM and SLF compounds as when PM enters in contact with the epithelial lung fluid (Calas et al., 2017) and then

direct comparison may not be fully accurate.

    The method used in this study gives very robust results and is promising for practical application. However, since it has some limitations, we hereafter list some possible improvements. First, as previously discussed, the model is strongly constrained by the explanatory variable, which are the PM sources contributions obtained with a PMF analysis. The PMF model has uncertainties of two different natures, inherent to the model: 1) mathematical uncertainties on the sources contributions and

2) frequent mixing profiles, due to co-linearity induced, e.g., mainly by meteorology. In our study, we might encounter such

mixing for the biogenic sources. An improvement would be to bootstrap the PMF results and use these uncertainties in the OP inversion in order to see its sensitivity.

Even if it has been shown that mainly $PM_{2.5}$ deposit in lung alveoli (Fang et al., 2017), $PM_{10}$ are still a public health concern and under regulation in EU and France. $PM_{10}$ has the advantage to encompass all parts of PM potentially reaching the lower respiratory track. However, in doing so, a source of uncertainty probably arises from the mixing, in our measurements systems, of PM populations with different chemical characteristics (i.e. acidity), that can influence the OP (i.e. changing solubility of trace metal, for example).This potential artifact, already existing for $PM_{2.5}$, may be reinforced with $PM_{10}$.

Another debatable choice is setting the intrinsic OP to zero for the source with a negative intrinsic OP during the stepwise regression process. Some chemical species may act as anti-oxidants which lead to "negative" intrinsic OP for the associated PM source. Namely, the polyols from the primary biogenic source, that include species like mannitol, are known to present strong anti-oxidant capabilities (Liu et al., 2010) and bacteria can halve the OP of copper-rich PM (Samake et al., 2017). Further studies should focus on this topic in order to better understand this potential effect.

Other choices of targets for optimization, and of penalty functions to promote the positivity of the coefficients, are possible. However, we think that our proposals manage to strike a balance between a satisfactory handling of the uncertainties of the problem and ease of application using existing statistical frameworks.

## 5  Conclusions

Based on one-year $PM_{10}$ sampling at an urban site located in Chamonix (France) associated with chemical speciation and Oxidative Potential (OP) measurements with the DTT and OP AA assays, we successfully established a method to attribute the contribution of the PM sources to the observed OP. The main conclusions of this study are summarized hereafter:

1. The different sources present different OP AA and OP DTT per microgram of PM with intrinsic OP differences between sources up to a factor of 20.

2. The biomass burning and vehicular sources seem to be the leading sources of the OP $AA_v$ and OP $DTT_v$ in Chamonix. On an annual basis, they represent together 78 % of the OP $AA_v$ and 54 % of the OP $DTT_v$ apportionment.

3. The two OP assays present different view on the PM sources based on their specific chemical selectivity as illustrated by the salt source that does not contribute to the OP $AA_v$ but to the OP $DTT_v$.

4. The relative mass contributions of the sources to the $PM_{10}$ differ from their relative OP $AA_v$ and OP $DTT_v$ contributions. For instance, the vehicular source has a larger contribution to the total OP AA and OP DTT than to the total $PM_{10}$ mass, whereas the sulfate rich source appears to be a minor source of OP $AA_v$ but an important source of PM mass. If OP is a proper metric of health impact of PM on population, the PM mass is not fully appropriate for PM regulations targeting public health.

Finally, even if OP metric is correlated to health outcomes, this study cannot directly attribute toxicity to one source or another. Is sporadic exposure to PM with high OP values or chronical exposure to PM with low OP values sufficient to

provoke health damage? As the DTT and AA tests point different sources as the main ROS-generating, does one of them is the more linked to toxicological effects (if any)? To answer these questions, more cross-over studies involving OP measurements, epidemiology and toxicology are needed.

*Competing interests.*  The authors declare no conflict of intersest or competing financial interests.

5  *Acknowledgements.*  This work was funded in part by Primequal (DECOMBIO program in the Arve Valley, grant ADEME 1362C0028) and by ANSES (ExPOSURE program, grant 2016-CRD-31). The funding of the PhD for S. Weber is provided by the Ecole Normale Supérieure. The Région Auvergne Rhône-Alpes funded the PhD grant for F. Chevrier. The Université Grenoble Alpes funded the PhD grant of A. Calas with a Président Award. The funding of the post-doctoral position for D. Salameh comes from the SOURCES program (ADEME Grant 1462C0044). This study was also supported by direct funding by IGE and LCME (technician salary), the LEFE CHAT Potentiel oxydant program and the LABEX OSUG@2020 (ANR-10-LABX-56) (both for funding analytical instruments). ATMO AuRA conducted all the logistical aspect of the sample collection in the field.

The authors would like to thanks Lisa Fluchaire, Jean-Charles Francony, Coralie Conniès, Vincent Lucaire, and Fanny Masson for their dedicated work for the samples analyses, together with many people from Atmo AuRA for collection of samples in the field. Many thanks also to Jesus Carrete Montaña for fruitfully improving the ideas in this work.

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
