# Peer review of "An apportionment method for the Oxidative Potential to the atmospheric PM sources: application to a one-year study in Chamonix, France."

_Atmospheric Chemistry and Physics, 2017_

## Referee Comment (RC1) · Anonymous Referee #1 · 24 Jan 2018

This paper uses a PMF analysis to determine the sources of PM10 OP measured with two assays, ascorbic acid (AA) and dithiothreitol (DTT). A multiple regression analysis is then used to derive a linear model to predict the OP for each assay with sources as the independent variable. This model provides the intrinsic OP (OP/mass) for each source. Relative contributions of the sources to PM10 mass and the OP for each assay are contrasted. The results are interesting and add more general insight into these assays. However, there are a number of issues. 1) The PMF/multiple regression analysis approach on OP has been done before (Bates et al). This should be recognized and

more comparisons between this and the Bates method provided. This is not a novel idea, as seems to be indicated in the Abstract. 2) After noting that a current issue with various studies investigating OP of aerosols is the lack of a standard method, the authors utilize a non-standard method for the OP assay. This makes comparisons with other reported studies difficult. 3) The authors use PM10 in the analysis, which leads to further difficulties in comparing with other studies (many are PM2.5) and makes the source apportionment much more complicated. Furthermore, it raises the possibility of greater artifacts arising in the OP analysis due to interactions between aerosol chemical species that are not mixed in the ambient aerosol, or likely to be mixed when deposited in the respiratory system. Although this adds to the confusion when comparting OP results, the results are of sufficient interest for publication in this journal. Specific comments follow.

Specific Comments:

Pg 2 Line 1. Specifically explain how OP measurements relies on surface area and particle size

Pg 2 lines 17 to 25: Issues with PMF. This paragraph is rather opaque and possibly unnecessary. First, PMF analysis of aerosol data sets is now very common practice, so many of the issues have been addressed. Thus it is not clear why such detailed discussions are included. It would be better to refer to some of the original papers. Furthermore, a number of investigators have reported using PMF on OP data. These should also be cited. A better literature review by the authors is warranted.

Pg 4 lines 19 to 21. Explain specifically how meteorology and inversions increase correlations between measured species. Take inversions, for example, how is co-variability affected by a factor that changes concentrations of all independent variables by roughly the same amount?

Pg 4 line 32, what does geochemically sounded mean?

[Figure]

Pg 5, what does pulled up maximally mean?

Was the DTT assay performed in simulated lung fluid (SLF)? If so, this deviates from the standard DTT protocol as described by Cho et al, 2005. This means the DTT activities reported in this work cannot necessarily be compared to most other published work, which follow the Cho method. This issue is important since it potentially adds confusion to the public literature on OP. This should be made very clear in the Abstract, Conclusions etc. Optimally, a conversion factor comparing this SLF_DTT to the standard method DTT activities would be included in the paper to help in interpreting the results. It is rather surprising these authors have done this given statements in the paper on limitations with no standard method used in practice (ie, the statement is made, then a non-standard practice is utilized).

Pg 5 Section 2.4 Does the filter extraction method include only water soluble species or both water and water insoluble species. For example, does the method extract BC? It is later stated that BC is correlated with the measured OP, but it is unclear if the assays are actually exposed to BC. More details on the extraction method are needed.

No discussion given on blanks, which are critical in OP measurements. Is the OP data blank corrected?

Pg 5 line 26, the unit nmol min-1 m-1 is incorrect.

Pg 6 line 19, typo, has shows. . .

Pg 6 line 20 and 21. What does the following line mean? . . . This underlines that the assays are sensitive to different ROS. . .. Is it being asserted that the assays are measuring ROS on the particle? This is not what these assays are designed to measure. I think the line gives a false impression, versus what the authors really mean.

Fig 10. The vehicular source does not appear constant throughout the year. The mean is very likely higher during cold seasons versus warm seasons, contrary to what is stated in the paper. Any explanation?

How is the multiple regression analysis performed in this paper novel relative to the published results of Bates et al (which is cited)? A detailed contrast would be valuable since it appears this paper is following, in general, the same approach as Bates et al. Or maybe it differs?

Pg 8, line 35. Why does the OP never display negative values? Is it because they were thrown out in the data set, ie samples with low masses where not included. These would be cases where the measurements would be near the detection limits and negative values possible. This needs more explanation as it can bias the results.

Pg 9, line 1. A non-zero intercept is also possible due to sampling artifacts associated with the filter sampling. Un-denuded sampling onto quartz filters are known to have positive artifacts. Add a discussion on possible effects of filter artifacts on this analysis.

Pg 10 line 13 and 14. These assays do not measure the ability to generate ROS, they measure the depletion of the antioxidant. There is a big difference.

Pg 10 Regarding the discussion comparing the intrinsic OP levels. Why not compare to other reported intrinsic values? For DTT, see for example, Shiraiwa et al (Shiraiwa, M., K. Ueda, A. Pozzer, G. Lammel, C. J. Kampf, A. Fushimi, S. Enami, A. M. Arangio, J. Frohlich-Nowoisky, Y. Fujitani, A. Furuyama, P. S. J. Lakey, J. Lelieveld, K. Lucas, Y. Morino, U. Poschl, S. Takahama, A. Takami, H. Tong, B. Weber, A. Yoshino, and K. Sato (2017), Aerosol health effects from molecular to global scales, Envir. Sci Technol. , 51, 13545-13567). Keep in mind that much data is likely PM2.5.

Pg 10. When comparing the results of this study to others, eg, Bates, et al., Fang et al., Verma et al., Charrier et al., keep in mind that those studies were PM2.5 not PM10, which could have a significant effect. Furthermore, the assay was performed differently in this study.

Pg 13 line 17. In what sense are the results very good. This is an opinion. Are the results very good because the models could reproduce the total observed OPs?

Uncertainties. A large uncertainty not considered is mixing aerosols over a broad size range into a single liquid sample and testing the OP of that mixture. These particles are definitely not internally mixed in the ambient atmosphere, nor in contact when deposited in the lung. Using PM10 makes this situation much worse as it mixes aerosol of widely different sources (secondary and mechanically generated primary, for fine and coarse, respectively). A number of papers show that there are both antagonistic and synergistic interactions possible between species that will affect the OP measurement (eg, see: Xiong, Q., H. Yu, R. Wang, J. Wei, and V. Verma (2017), Rethinking The Dithiothreitol (DTT) Based PM Oxidative Potential: Measuring DTT Consumption versus ROS Generation, Envir. Sci. Technol, 51, 6507-6514)

The last line of the paper needs to be edited.
* * *

---

## Author Comment (AC1) · 19 Mar 2018

This paper uses a PMF analysis to determine the sources of PM10 OP measured with two assays, ascorbic acid (AA) and dithiothreitol (DTT). A multiple regression analysis

is then used to derive a linear model to predict the OP for each assay with sources as the independent variable. This model provides the intrinsic OP (OP/mass) for each source. Relative contributions of the sources to PM10 mass and the OP for each assay are contrasted. The results are interesting and add more general insight into these assays. However, there are a number of issues. 1) The PMF/multiple regression analysis approach on OP has been done before (Bates et al). This should be recognized and more comparisons between this and the Bates method provided. This is not a novel idea, as seems to be indicated in the Abstract. 2) After noting that a current issue with various studies investigating OP of aerosols is the lack of a standard method, the authors utilize a non-standard method for the OP assay. This makes comparisons with other reported studies difficult. 3) The authors use PM10 in the analysis, which leads to further difficulties in comparing with other studies (many are PM2.5) and makes the source apportionment much more complicated. Furthermore, it raises the possibility of greater artifacts arising in the OP analysis due to interactions between aerosol chemical species that are not mixed in the ambient aerosol, or likely to be mixed when deposited in the respiratory system. Although this adds to the confusion when comparting OP results, the results are of sufficient interest for publication in this journal. Specific comments follow. Authors thank this referee for his useful review. We tried out to answer him in the following discussion.

Pg 2 Line 1. Specifically explain how OP measurements relies on surface area and particle size The OP is a measure of reactivity between some antioxidants or surrogates (AA or DTT) and the PM. The reactivity is function of the chemistry and the number of reaction site. The particle size depends on several factors, along them the chemistry and the lifetime PM (Gehling and Dellinger, 2013). Furthermore, the higher the surface area is, the more likely the number of reaction site is. Surface area and particle size regarding OP measurements was also addressed specifically by (Sauvain et al., 2013). These references were added in the manuscript.

Gehling, W., & Dellinger, B. (2013). Environmentally persistent free radicals and their

lifetimes in PM2. 5. Environmental science & technology, 47(15), 8172-8178. Sauvain, J.-J., Rossi, M.J., Riediker, M., 2013. Comparison of three acellular tests for assessing the oxidation potential of nanomaterials. Aerosol Science and Technology 47, 218-227.

Pg 2 lines 17 to 25: Issues with PMF. This paragraph is rather opaque and possibly unnecessary. First, PMF analysis of aerosol data sets is now very common practice, so many of the issues have been addressed. Thus it is not clear why such detailed discussions are included. It would be better to refer to some of the original papers. Furthermore, a number of investigators have reported using PMF on OP data. These should also be cited. A better literature review by the authors is warranted. In this paragraph we explain why we didn't estimate the OP with the chemical species as explanatory variable but with the PMF sources instead. We explain what are the possible biases if we assess the linear regression directly with the chemical species and how using sources' contribution instead mitigate them. We modified the text as follows:

Another option is to consider the sources contribution instead of the chemical species (Bates et al., 2015; Fang et al., 2015, 2016; Verma et al., 2014). Indeed, working directly with chemical species involves assessing an exhaustive composition characterization. This is rather impossible since many species in the complex mixture of aerosols remain unidentified. Moreover, if a detailed composition (which can sometimes include up to 150 species (Waked et al., 2014)) is provided, at least the same number of samples for OP measurements is needed, otherwise, the system remains underdetermined. Reducing the system by direct truncation is not possible since species contributing to OP could be dropped, inducing some degree of unknown bias. Conversely, if the explanatory variables are the sources' contributions, biases are mitigated. However, the sources dynamics need to be determined for a long period of time in order to reflect the climatology of the location. Moreover, the composition of a given named source may vary according to its location (Belis et al., 2013). To mitigate these issues, we decide to use a PMF approach instead of a CMB model to better render the local specificities of the sources. Indeed, the CMB averages the sources profil's from different studies and is then locally biased. Furthermore, in this study a whole year of analysis is used as input of the PMF. We then have a climatological view of the sources dynamics.

Bates, J. T., Weber, R. J., Abrams, J., Verma, V., Fang, T., Klein, M., Strickland, M. J., Sarnat, S. E., Chang, H. H., Mulholland, J. A., Tolbert, P. E. and Russell, A. G.: Reactive oxygen species generation linked to sources of atmospheric particulate matter and cardiorespiratory effects, Environ. Sci. Technol., 49(22), 13605–13612, doi:10.1021/acs.est.5b02967, 2015. Belis, C. A., Karagulian, F., Larsen, B. R. and Hopke, P. K.: Critical review and meta-analysis of ambient particulate matter source apportionment using receptor models in Europe, Atmos. Environ., 69, 94–108, doi:10.1016/j.atmosenv.2012.11.009, 2013. Fang, T., Guo, H., Verma, V., Peltier, R. E. and Weber, R. J.: PM2.5 water-soluble elements in the southeastern United States: automated analytical method development, spatiotemporal distributions, source apportionment, and implications for heath studies, Atmospheric Chem. Phys., 15(20), 11667–11682, doi:10.5194/acp-15-11667-2015, 2015. Fang, T., Verma, V., Bates, J. T., Abrams, J., Klein, M., Strickland, M. J., Sarnat, S. E., Chang, H. H., Mulholland, J. A., Tolbert, P. E., Russell, A. G. and Weber, R. J.: Oxidative potential of ambient water-soluble PM 2.5 in the southeastern United States: contrasts in sources and health associations between ascorbic acid (AA) and dithiothreitol (DTT) assays, Atmospheric Chem. Phys., 16(6), 3865–3879, doi:10.5194/acp-16-3865-2016, 2016. Verma, V., Fang, T., Guo, H., King, L., Bates, J. T., Peltier, R. E., Edgerton, E., Russell, A. G. and Weber, R. J.: Reactive oxygen species associated with water-soluble PM2 in the southeastern United States: spatiotemporal trends and source apportionment., Atmospheric Chem. Phys., 14(23), 12915–12930, doi:10.5194/acp-14-12915-2014, 2014. Pg 4 lines 19 to 21. Explain specifically how meteorology and inversions increase correlations between measured species. Take inversions, for example, how is co-variability affected by a factor that changes concentrations of all independent variables by roughly the same amount? We add in the manuscript the following explanation:

"During temperature inversion in Alpine valley, pollutants are stuck into the Atmospheric Boundary Layer (ABL) and cannot be removed by wind. Such inversion may be stable during several days. As a result, the different emission sources during that period of time add together and the dynamic from the different sources is masked. In other word, one sample does not integrate anymore only emissions during the sampling time, but also emissions of the previous days. This end-up with chemical species in one sample that should not be present together, respect to the temporality of their respective sources. Thereby, their correlation is increased".

Pg 4 line 32, what does geochemically sounded mean? We didn't want to constraint to much the PMF. We then decided to put a minimal amount of constraint based on our prior knowledge of the sources. We modified our sentence as follows:

A minimal set of constraints based on prior and external geochemical knowledge of sources fingerprints was applied

Pg 5, what does pulled up maximally mean? "Pull up" refer to an option of the EPA PMF5.0 software: • Pull up try to increase the amount of the desired specie in the given factor • Pull down does the opposite. Shortly, the new solution of the PMF differs from the previous one from a certain amount dQ, and the PMF software let use choose the strength of the variation (Pull up / pull up maximally). For further details, refer to the EPA PMF5.0 userguide. We modify the following explanation in the text:

• in the biomass burning factor, the contributions of levoglucosan, potassium, methoxyphenols and Bcbb were increased, whereas the Bcff and HOP were set to 0, • HOP was increased in the vehicular factor. We increased the concentration of the species in the factors thanks to the"pull up maximally" option of the EPA PMF5.0 software (US EPA, 2017), which try to increase the contribution of the given specie to the factor.

Was the DTT assay performed in simulated lung fluid (SLF)? If so, this deviates from the standard DTT protocol as described by Cho et al, 2005. This means the DTT activities reported in this work cannot necessarily be compared to most other published work, which follow the Cho method. This issue is important since it potentially adds confusion to the public literature on OP. This should be made very clear in the Abstract, Conclusions etc. Optimally, a conversion factor comparing this SLF_DTT to the standard method DTT activities would be included in the paper to help in interpreting the results. It is rather surprising these authors have done this given statements in the paper on limitations with no standard method used in practice (ie, the statement is made, then a non-standard practice is utilized).

DTT assay was performed after a PM extraction in SLF according to Calas et al., 2017. As shown by Calas et al., 2017 OP results after SLF extraction diverge by a little % with regards to a water extraction as done by Cho et al., 2005. Our protocol post-extraction is performed according to Charrier et al., 2012 which have modified Cho et al., 2005, carried out Chelex-treated phosphate buffer to remove trace metal contamination. In fact, Cho et al., (according to Kumagai et al., 2022) used EDTA to purify the phosphate buffer leading to further metal effect inhibition during the assay. Finally, the percentage or conversion factor is not predictable between OP(SLF_DTT) and OP DTT since the extraction in SLF is mimicking the contact between particles and lung epithelium and some components might be complexed or not as in physiologic conditions. We clarified the use of this protocol in the abstract and possible limitations of the study

We add the following limitation:

First of all, when comparing with others previous studies we should note that our PM extraction of samples were done in a SLF and not in water. This induces a difference of several percent in OP measurement which is not predictable for the complexes occurring between PM and SLF compounds as when PM enters in contact with the epithelial lung fluid (Calas et al., 2017), and then direct comparison may not be fully accurate.

Pg 5 Section 2.4 Does the filter extraction method include only water soluble species or both water and water insoluble species. For example, does the method extract BC? It

is later stated that BC is correlated with the measured OP, but it is unclear if the assays are actually exposed to BC. More details on the extraction method are needed.

We agree with the reviewer and added this sentence to clarify our protocol:

The filter extraction method includes both water soluble and insoluble species. After the SLF extraction, particles removed from filter are not filtrated; the whole extract is injected in the multiwall plate.

No discussion given on blanks, which are critical in OP measurements. Is the OP data blank corrected? Three filter blanks are included in every plate (OP AA and OP DTT) of the protocol. The value of the blank are then deduced from the sample measurement. This was added in the methodology section:

Three filter blanks are included in every plate (OP AA and OP DTT) of the protocol. The value of the blank are then deduced from the sample measurement.

Pg 5 line 26, the unit nmol min-1 m-1 is incorrect. Modified accordingly nmol min-1m-3. Thanks for your attentive review.

Pg 6 line 19, typo, has shows. . . Thanks. Corrected to "shows".

Pg 6 line 20 and 21. What does the following line mean? . . . This underlines that the assays are sensitive to different ROS. . .. Is it being asserted that the assays are measuring ROS on the particle? This is not what these assays are designed to measure. I think the line gives a false impression, versus what the authors really mean. We apologize for the misunderstanding, DTT and AA assays are indirect ROS measurements. We corrected our sentence as follows:

This underlines that the assays are sensitive to different chemicals species carried out by PM.

Fig 10. The vehicular source does not appear constant throughout the year. The mean is very likely higher during cold seasons versus warm seasons, contrary to what

is stated in the paper Any explanation? In Alpine valley some strong and persistent inversion layer occurs during winter. This leads to accumulation of pollutant in the ABL. As a result, even if the emission is constant, the concentration of the vehicular source increases during winter. We modify the text as follows:

The vehicular source is quite constant all over the year. Indeed, the higher concentration during winter may be attributed to accumulation in the ABL, and not to an increase of emission.

How is the multiple regression analysis performed in this paper novel relative to the published results of Bates et al (which is cited)? A detailed contrast would be valuable since it appears this paper is following, in general, the same approach as Bates et al. Or maybe it differs? Indeed, we use a quite similar approach as there is not so many way to perform a multiple linear regression. However, there are several points where we differ. First of all, in the backward elimination, Bates et al decided to base their rejection criteria on the p-value. We decided to remove a source based on its negativity. Indeed, we may have a statistically significant negative value. But according to us, a source with negative intrinsic OP does not have a geo-chemical sens as the air is known to be a strong oxidant milieu. Secondly, the approach of Bates et al, uses ordinary least square (OLS). However, as we have an estimation of our measurements uncertainty, we use a weighted least square (WLS) regression. Finally, we propose a way to estimate the uncertainty of our estimated OP thanks to a Monte-Carlo method, which is not provided in the previous study. Moreover, the method proposed here does not only include the multiple linear regression (MLR) but also the use of the PMF model instead of the CMB one. Indeed, the MLR is highly sensitive to the explanatory variable and we decide to use the local sources' profile (PMF) instead of the chemical mass balance method with ensemble-averaged source impact profiles.

We add the following paragraph to explain the differences between the two methods:

The method proposed here is an improvement of the one of Bates et al (2015) and our

methods differ in several points. First of all, our backward elimination criteria is based on the negativity of a source and not in its p-value. Indeed, a source might present a statistically significant negative value. But according to us, a source with negative intrinsic OP does not have a geo-chemical sense as the air is known to be a strong oxidant milieu. Secondly, as Bates et al (2015) didn't measure the uncertainty of their OP samples, they used an ordinary least square (OLS) regression. On the opposite, we have an estimation of our measurements uncertainty thanks to triplicate. We then use a weighted least square (WLS) regression instead. Finally, we propose a way to estimate uncertainties of our estimated OP with a Monte-Carlo method, which is not provided in the previous study. Moreover, the method proposed here does not only include the multiple linear regression (MLR) but also the use of the PMF model instead of the CMB one. Indeed, the MLR is highly sensitive to the explanatory variable and we decide to use the local sources' profile (PMF) instead of the chemical mass balance method with ensemble-averaged source impact profiles.

Pg 8, line 35. Why does the OP never display negative values? Is it because they were thrown out in the data set, ie samples with low masses where not included. These would be cases where the measurements would be near the detection limits and negative values possible. This needs more explanation as it can bias the results. Some samples were not analyzed due to the too low PM mass impacted in the filters. In all analyzed samples, the OP (AA and DTT) never displays negative value. This is also true for the samples with the lowest PM mass. Indeed, days with low mass are not necessary day with low OP or near detection limit. We discarded these days only to have comparable data, based on the same protocol for the extraction and measurement (since DTT assay can display non-linear response when you are not extracting at iso-mass concentration (Charrier et al., 2016)) . As a result, we don't think that the exclusion of days with too low mass induces a bias in the OP and could hide the possible negative OP for these days. As far as we know, the OP of these days may be high, low or negative. But of course, as we did not measure these sample, this is only an assumption.

Pg 9, line 1. A non-zero intercept is also possible due to sampling artifacts associated with the filter sampling. Un-denuded sampling onto quartz filters are known to have positive artifacts. Add a discussion on possible effects of filter artifacts on this analysis.

We are not sure to fully understand the reviewer as the question seems to mix chemistry-related and OP-related issues. However, we definitely agree that impacts of the sampling artifacts is a very important topic, most probably for both aspects. Sampling artifacts of PM with filters is well known. However, to the best of our knowledge, no previous study focused on the specific point of how it impacts PMF studies. This is indeed an intricated issue since PMF requires chemical profiles of sources which are rather stable in time, while sampling artifacts are by nature dependent upon the pressure and temperature of sampling. We may point out that Favez et al. (2010) (among other studies) did source apportionment inter-comparison between off-line samples (filters) and AMS sampling, and found a relatively good agreement between the different techniques that are not prone to sampling artifacts in the same way. Therefore, the impact of artifact may be mitigated for source apportionment.

When it comes to the impact of sampling artifacts on OP measurements, it also has been shown recently that heterogeneous reactions may happen between the gaseous oxidant (ozone, free radical) and the surface of the filter (Malynuk et al., 2015). Such reaction may have an impact on the OP measurements from filter sampling. However, again, there is no specific work presented in the literature on the direct impact of the change in chemistry linked to sampling artefact on the OP measurements.

Nevertheless, the paragraph of our article quoted for this remark do not refer to this topics. We compared OP measured and OP modeled and compute the linear regression between them. Therefore, we work on the "on-filter OP", and it does not seem a point to discuss sampling artifact in this context. But we fully agree with the reviewer that OP measured on the filter may not reflect actual OP in the real atmosphere. Some in depth study related to OP of semi-volatile species are needed to estimate the "sampling artifact of OP". Again, to the best of our knowledge, no study was focused in this

specific issue yet, and our data do not allow such investigations.

Melymuk, L., Bohlin-Nizzetto, P., Prokeš, R., Kukučka, P. and Klánová, J.: Sampling artifacts in active air sampling of semivolatile organic contaminants: Comparing theoretical and measured artifacts and evaluating implications for monitoring networks, Environmental Pollution, 217, 97 106, doi:10.1016/j.envpol.2015.12.015, 2016.

Favez, O., El Haddad, I., Piot, C., Boréave, A., Abidi, E., Marchand, N., Jaffrezo, J.-L., Besombes, J.-L., Personnaz, M.-B., Sciare, J., Wortham, H., George, C. and D Anna, B.: Inter-comparison of source apportionment models for the estimation of wood burning aerosols during wintertime in an Alpine city (Grenoble, France), Atmospheric Chemistry and Physics, 10(12), 5295 5314, doi:10.5194/acp-10-5295-2010, 2010.

Pg 10 line 13 and 14. These assays do not measure the ability to generate ROS, they measure the depletion of the antioxidant. There is a big difference. We apologize for the confusion and perfectly agree. This is a big shortcut that does not take into account the numerous chemical reactions that might occur. We rephrase the following sentence: The various sources do not have the same ability to generate ROS.

By: The various sources do not have the same reactivity toward the AA and DTT.

Pg 10 Regarding the discussion comparing the intrinsic OP levels. Why not compare to other reported intrinsic values? For DTT, see for example, Shiraiwa et al (Shiraiwa, M., K. Ueda, A. Pozzer, G. Lammel, C. J. Kampf, A. Fushimi, S. Enami, A. M. Arangio, J. Frohlich-Nowoisky, Y. Fujitani, A. Furuyama, P. S. J. Lakey, J. Lelieveld, K. Lucas, Y. Morino, U. Poschl, S. Takahama, A. Takami, H. Tong, B. Weber, A. Yoshino, and K. Sato (2017), Aerosol health effects from molecular to global scales, Envir. Sci Technol. , 51, 13545-13567). Keep in mind that much data is likely PM2.5. Pg 10. When comparing the results of this study to others, eg, Bates, et al., Fang et al., Verma et al., Charrier et al., keep in mind that those studies were PM2.5 not PM10, which could have a significant effect. Furthermore, the assay was performed differently in this study.

[Figure]

This study is intended to be focused on the methodology and provides only a study case. We didn't aimed at going into too much interpretation of the Chamonix results. However, a incoming study with other sites will compare our results to previous studies in much more details. Also, as you pointed out, our study is on PM10 and with a singular extraction protocol. We then should be very cautious when comparing the intrinsic OP levels with other study as it was already reported that the OP may vary or not within the size of the PM (Gali et al., 2017; Styszko et al., 2017, Samara, 2017)

Gali, N. K., Jiang, S. Y., Yang, F., Sun, L. and Ning, Z.: Redox characteristics of size-segregated PM from different public transport microenvironments in Hong Kong, Air Qual. Atmosphere Health, 10(7), 833–844, doi:10.1007/s11869-017-0473-0, 2017. Styszko, K., Samek, L., Szramowiat, K., Korzeniewska, A., Kubisty, K., Rakoczy-Lelek, R., Kistler, M. and Giebl, A. K.: Oxidative potential of PM10 and PM2.5 collected at high air pollution site related to chemical composition: Krakow case study, Air Qual. Atmosphere Health, 10(9), 1123–1137, doi:10.1007/s11869-017-0499-3, 2017.

Samara, C.: On the Redox Activity of Urban Aerosol Particles: Implications for Size Distribution and Relationships with Organic Aerosol Components, Atmosphere, 8(10), 205, doi:10.3390/atmos8100205, 2017.

Pg 13 line 17. In what sense are the results very good. This is an opinion. Are the results very good because the models could reproduce the total observed Ops? "Very good" refers to the statistical result of our inversion procedure: normalized residual distribution centered in 0, $r^2$, intercepts, uncertainties. We replaced very good by very robust in the manuscript.

Uncertainties. A large uncertainty not considered is mixing aerosols over a broad size range into a single liquid sample and testing the OP of that mixture. These particles are definitely not internally mixed in the ambient atmosphere, nor in contact when deposited in the lung. Using PM10 makes this situation much worse as it mixes aerosol of widely different sources (secondary and mechanically generated primary, for fine

and coarse, respectively). A number of papers show that there are both antagonistic and synergistic interactions possible between species that will affect the OP measurement (eg, see: Xiong, Q., H. Yu, R. Wang, J. Wei, and V. Verma (2017), Rethinking The Dithiothreitol (DTT) Based PM Oxidative Potential: Measuring DTT Consumption versus ROS Generation, Envir. Sci. Technol, 51, 6507-6514) We perfectly agree. However, all size of particle may have health impact and then should be accounted. Moreover, a part of the coarse particle are in between the PM2.5 and PM10 (notably metals (Gietl et al., 2010)). The distinction PM10/PM2.5 and PM2.5/PM0.1 are norms and may not be fully physically based. On top of the different size, we also consider a temporal contribution constant over the day although we know that the dynamic of some sources may vary during one day. For instance Biomass Burning is more active during evening/night and the vehicular in the morning. As a consequence, their PM may not be internally mixed neither. We will emphasize this limitation as follows:

"An important source of uncertainty lies on the internal mixing of the aerosols as we integrate size of PM and temporal contribution of the sources. We average their contributions on a daily basis although we know that they occur at different hours of the day and with different size. As a consequence, their PM may not be internally mixed."

Gietl, J. K., Lawrence, R., Thorpe, A. J. and Harrison, R. M.: Identification of brake wear particles and derivation of a quantitative tracer for brake dust at a major road, Atmospheric Environment, 44(2), 141–146, doi:10.1016/j.atmosenv.2009.10.016, 2010.

The last line of the paper needs to be edited. "To answer these questions, more crossover studies including OP-epidemiology and toxicology are needed." We edited this line as follows:

"To answer these questions, more cross-over studies involving OP measurements, epidemiology and toxicology are needed."

Please also note the supplement to this comment:

[Figure]

https://www.atmos-chem-phys-discuss.net/acp-2017-1053/acp-2017-1053-AC1-supplement.pdf

**Supplement:**

[revised manuscript text omitted]

---

## Referee Comment (RC2) · Anonymous Referee #2 · 21 Mar 2018

The manuscript of Weber et al. represents the OP results obtained by analyzing a series of filter PM10 samples collected during a year-long period at an urban location in France, using two different assays, namely the dithiothreitol assay (DTT) and the ascorbic acid assay (AA). Combining results obtained by different analyses of the collected filters, including soluble ions, metals, PAHs and combining these results with PMF and linear regressions analyses for the identification of different sources and the subsequent attribution of redox-activity to different PM sources. It occurs that a large part of the observed OP is linked to biomass burning and vehicular sources for both

assays.

The paper is well written and easy to follow, though there are some issues and more thorough discussion should be made in specific sections. A very interesting point of the study is that the used assays appear to be sensitive to different ROS. Other than that the paper can be recommended for publication after addressing the issues listed below.

Specific comments:

1) Samples consist of PM10 while PM2.5 is most commonly used as being able to penetrate inside the respiratory system. Although the used range (PM10) surely covers the totality of the OP distribution, the difference of acidity between fine and coarse fraction surely plays a key role in the aerosol OP, influencing the solubility of metals (e.g. Fang et al. 2017). Authors should comment on this.

2) It is stated that the current study uses simulated lung fluid (SLF) solution, complicating the direct comparison with other studies. It should be clearly stated in the abstract and conlcusions section that a method different than the standard DTT protocol is used in order to avoid confusion. Furthermore, as seen in Calas et al. (2017), the OPDTT measured in Milli-Q water and three different SLF extracts does not present statistically significant differences. Authors should comment on the choice of extract. Finally, in the extraction phase (P5,L13) is different extraction volume used for different samples or only a different area of the used filter? This is not clear.

3) There is no mention of the LOD for the specific assays using the SLF, nor blank/ blank corrections.

4) When presenting the concentrations of the PMF sources, emphasis is only given for the correlation of OP solely with biomass burning and vehicular sources, even though it appears that "nitrate rich" source could also be correlated, as during winter enhanced nitrate concentrations are usually associated with biomass burning. Although mentioned further on (P12, L10) it should also be mentioned and commented on, here.

5) A more thorough discussion should be made in the Intrinsic OP section, namely a comparison with other values found in the literature (even though the majority concerns PM2.5) and the use or not of an intercept in the linear regression model. Furthermore, it is stated that other studies also highlight the importance of the vehicular source to explain the OP. In Verma et al. (2015) even though HOA (representing traffic) correlates significantly with OP at some sites, the generated linear regression models do never include HOA, though in some cases the linear regression model include copper. It is known that copper may originate from brake wearing, but also it can be linked to other anthropogenic activities, such as industry and/or coal burning.

Technical corrections:

Title: "Oxydative" should be corrected to "Oxidative"

Abstract, L1: "…induces cellular oxidative stress in vivo, leading to adverse…"

P6, L19: "…DTTv shows larger values…" (delete "has")

P7, L7: "…sources appear to be strongly correlated…"

P14, L18: "...biomass burning and vehicular sources…"

References

Vishal Verma, Ting Fang, Lu Xu, Richard E. Peltier, Armistead G. Russell, Nga Lee Ng, and Rodney J. Weber: Organic Aerosols Associated with the Generation of Reactive Oxygen Species (ROS) by Water-Soluble PM2.5, Environmental Science & Technology 2015 49 (7), 4646-4656, DOI: 10.1021/es505577w

Ting Fang, Hongyu Guo, Linghan Zeng, Vishal Verma, Athanasios Nenes, and Rodney J. Weber: Highly Acidic Ambient Particles, Soluble Metals, and Oxidative Potential: A Link between Sulfate and Aerosol Toxicity, Environmental Science & Technology 2017 51 (5), 2611-2620, DOI: 10.1021/acs.est.6b06151

---

## Author Comment (AC2) · 18 Apr 2018

The manuscript of Weber et al. represents the OP results obtained by analyzing a series of filter PM10 samples collected during a year-long period at an urban location in France, using two different assays, namely the dithiothreitol assay (DTT) and the ascorbic acid assay (AA). Combining results obtained by different analyses of the collected filters, including soluble ions, metals, PAHs and combining these results with

[Figure]

PMF and linear regressions analyses for the identification of different sources and the subsequent attribution of redox-activity to different PM sources. It occurs that a large part of the observed OP is linked to biomass burning and vehicular sources for both assays. The paper is well written and easy to follow, though there are some issues and more thorough discussion should be made in specific sections. A very interesting point of the study is that the used assays appear to be sensitive to different ROS. Other than that the paper can be recommended for publication after addressing the issues listed below.

The authors would like to thank the second referee for his/her review and very useful comments that helps us to improve the paper. We tried to answer his/her question point by point in the following discussion.

1) Samples consist of PM10 while PM2.5 is most commonly used as being able to penetrate inside the respiratory system. Although the used range (PM10) surely covers the totality of the OP distribution, the difference of acidity between fine and coarse fraction surely plays a key role in the aerosol OP, influencing the solubility of metals (e.g. Fang et al. 2017). Authors should comment on this.

We indeed agree with the reviewer that a difference exist between PM10 and PM2.5, both in term of processes influencing the OP values (like changing pH values, hence solubility of some species, according to the size), and of particle size prone to deposition in the lungs. However, in EU and France, PM10 are under regulation, –not PM2,5, and are used as alert tool for health issues. Therefore, they need to be investigated with this respect. Further, as we answered to the first reviewer, the distinctions in PM10, PM2.5 etc, are norms and are generally not be fully physically based considering the large variability of the modal size distributions observed in actual environments. Fang et al. (2017) indeed showed clear differences both for deposition of OP in respiratory tracks and in OP activity depending on the size of PM; but the threshold vary between PM1.18 and PM3.2, which can already make great differences in the composition of the PM compared to the PM2.5 population. For instance, in the study by Fang et al

(2017), Cu2+ presents its mean mass distribution value for Dp = 2.5 $\mu$m, therefore half of Cu2+ is in the PM2,5 fraction, but half of it is in larger sizes. However, we added this remark as a limitation and edited the text as follows, including also our answer to the first reviewer' comment (p14):

Even if it has been shown that mainly PM2.5 deposit in lung alveoli (Fang et al., 2017), PM10 are still a public health concern and under regulation in EU and France. PM10 has the advantage to encompass all parts of PM potentially reaching the lower respiratory track. However, in doing so, a source of uncertainty probably arises from the mixing, in our measurements systems, of PM populations with different chemical characteristics (i.e. acidity), that can influence the OP (i.e. changing solubility of trace metal, for example).This potential artifact, already existing for PM2.5, may be reinforced with PM10.

Ting Fang, Hongyu Guo, Linghan Zeng, Vishal Verma, Athanasios Nenes, and Rodney J. Weber: Highly Acidic Ambient Particles, Soluble Metals, and Oxidative Potential: A Link between Sulfate and Aerosol Toxicity, Environmental Science & Technology 2017 51 (5), 2611-2620, DOI: 10.1021/acs.est.6b06151

2) It is stated that the current study uses simulated lung fluid (SLF) solution, complicating the direct comparison with other studies. It should be clearly stated in the abstract and conclusions section that a method different than the standard DTT protocol is used in order to avoid confusion. Furthermore, as seen in Calas et al. (2017), the OPDTT measured in Milli-Q water and three different SLF extracts does not present statistically significant differences. Authors should comment on the choice of extract. Finally, in the extraction phase (P5,L13) is different extraction volume used for different samples or only a different area of the used filter? This is not clear.

We agree that we should keep in mind the difference for the extraction protocol when trying to compare studies. This comment is also shared with the first reviewer and text modification is given in our answer. We clarified the extraction protocol in the methodology part and added a paragraph in the discussion concerning the comparativeness of our solution.

However, we disagree with the term "standard DTT protocol". To the best of our knowledge, there is still no consensus toward a standardized DTT protocol. Indeed, even if Cho et al. (2005) is a general reference, several protocols have evolved from this starting one, including changes that can make large differences: removing EDTA considering its chelating effect (Charrier and Anastasio, 2012); changing the temperature of reaction and the times intervals of measurement of the kinetics ("under linear condition (DTT loss < 20%)" in Cho et al., (2005), "0, 4, 13, 23, 30 and 41 min" in Fang et al., (2015), "0, 10, 20, 30, 40 and 50 min" in Jedynska et al.,( 2017), "0, 15 and 30 min" in Calas et al. (2017)) ; changing initial DTT concentrations: "160 $\mu$M" in Fujitani et al. (2017) ; "0.5 mM" in Calas et al. (2017), "1 mM" in Fang et al. (2015); "100 mM" in Jedynska et al. (2017)), etc. Moreover, it has been shown that the DTT response is not linear according to the amount of reacting species (i.e. the mass of PM) (Charrier et al., 2016), and it is definitively the main bias when comparing different studies using different PM mass extracted.

Calas et al (2018) showed that no statistical significant difference was found when using Gamble+DPPC or Milli-Q water as extraction fluids for the DTT the assay. However, in Calas et al (2018), only 5 samples were used and we clearly observed a small but constant higher OP in Milli-Q, that we explained by complexes that could take place between some species and the Gamble solution. Further, a significant difference was found between the extraction in Milli-Q water and in another lining fluid (the artificial lysosomal fluid); we did not selected this fluid for our standard protocol in our lab, since it is representative of inflammation in lung, and is therefore not relevant for all physiological conditions. As a result, we rather choose to keep the Gamble+DPPC protocol, as it is closer to biological conditions and may closely reflect the complexation occurring at the surface of lung epithelium when PM deposit.

Concerning the extraction phase, as all samples account for 24 h of sampling at constant flow rate (30 m$^3$/h), we adjust for each sample the surface of the filter needed in order to extract always the same mass of PM (selected in the range of linearity), has it has been shown that OPDTT and OPAA vary non-linearly with the mass of PM in assays. We clarify this in the text, as follows:

The extraction took place into SLF at iso-mass. All samples were analyzed at 10 $\mu$g.mL-1 of PM, by adjusting the area of filter extracted.

Calas, A., Uzu, G., Martins, J. M. F., Voisin, D., Spadini, L., Lacroix, T. and Jaffrezo, J.-L.: The importance of simulated lung fluid (SLF) extractions for a more relevant evaluation of the oxidative potential of particulate matter, Scientific Reports, 7(1), doi:10.1038/s41598-017-11979-3, 2017.

Fang, T., Verma, V., Guo, H., King, L. E., Edgerton, E. S. and Weber, R. J.: A semi-automated system for quantifying the oxidative potential of ambient particles in aqueous extracts using the dithiothreitol (DTT) assay: results from the Southeastern Center for Air Pollution and Epidemiology (SCAPE), Atmospheric Measurement Techniques, 8(1), 471 482, doi:10.5194/amt-8-471-2015, 2015.

Fujitani, Y., Furuyama, A., Tanabe, K. and Hirano, S.: Comparison of Oxidative Abilities of PM2.5 Collected at Traffic and Residential Sites in Japan. Contribution of Transition Metals and Primary and Secondary Aerosols, Aerosol and Air Quality Research, 17(2), 574 587, doi:10.4209/aaqr.2016.07.0291, 2017.

Charrier, J. G., McFall, A. S., Vu, K. K.-T., Baroi, J., Olea, C., Hasson, A. and Anastasio, C.: A bias in the mass-normalized DTT response An effect of non-linear concentration-response curves for copper and manganese, Atmospheric Environment, 144, 325 334, doi:10.1016/j.atmosenv.2016.08.071, 2016.

Charrier, J. G. and Anastasio, C.: On dithiothreitol (DTT) as a measure of oxidative potential for ambient particles: evidence for the importance of soluble transition metals, Atmospheric Chemistry and Physics Discussions, 12(5), 11317 11350,

doi:10.5194/acpd-12-11317-2012, 2012

Jedynska, A., Hoek, G., Wang, M., Yang, A., Eeftens, M., Cyrys, J., Keuken, M., Ampe, C., Beelen, R., Cesaroni, G., Forastiere, F., Cirach, M., de Hoogh, K., De Nazelle, A., Nystad, W., Akhlaghi, H. M., Declercq, C., Stempfelet, M., Eriksen, K. T., Dimakopoulou, K., Lanki, T., Meliefste, K., Nieuwenhuijsen, M., Yli-Tuomi, T., Raaschou-Nielsen, O., Janssen, N. A. H., Brunekreef, B. and Kooter, I. M.: Spatial variations and development of land use regression models of oxidative potential in ten European study areas, Atmospheric Environment, 150, 24 32, doi:10.1016/j.atmosenv.2016.11.029, 2017.

3) There is no mention of the LOD for the specific assays using the SLF, nor blank/blank corrections.

Indeed, this is perfectly right. The first reviewer also pointed it out. This was forgotten, and we added the following in the text :

Three filter blanks (laboratory blank) are included in every plate (OP AA and OP DTT) of the protocol. The average values of these blanks are then subtracted from the sample measurement of this plate. LOD value is defined as three times of the standard deviation of laboratory blanks measurements (blank filters in Gamble+DPPC solution).

4) When presenting the concentrations of the PMF sources, emphasis is only given for the correlation of OP solely with biomass burning and vehicular sources, even though it appears that "nitrate rich" source could also be correlated, as during winter enhanced nitrate concentrations are usually associated with biomass burning. Although mentioned further on (P12, L10) it should also be mentioned and commented on, here.

We choose to consider as "correlated" only the variables with r values above 0.6 (p<0,001 for n=85). Thus, the correlations between OP and the nitrate rich source (Fig. S3) are below this threshold of 0.6 (resp. 0.45 and 0.55 for OP AAv and OP DTTv). This is why we do not say that this factor is correlated with OP's. However, we agree
that this is an arbitrary threshold. When we discuss it later in the text (page 10, line 11), we do not discuss about the correlation coefficient between PMF source and OP, but instead about the intrinsic OP of each source. This "low" correlation OP/PMF source but relatively high OP contribution is one of the key point of the multiple linear regression. We see that highly correlated source may have "low" intrinsic OP DTT (namely biomass burning/OP DTTv r=0.80 but intrinsic OP $DTT_{BB}$=0.07 nmol/min/$\mu$g) and poorly correlated source may have high intrinsic OP (namely Secondary biogenic/OP DTT r=0.40, but intrinsic OP $DTT_{SOA}$=0.18 nmol/min/$\mu$g). This emphasize the need of a more sophisticated statistical tool than simple univariate correlation when dealing with sources contribution to the OP.

We added the modified paragraph in the discussion: p7, l8:

Briefly, the vehicular and biomass burning sources appear to be strongly correlated to both OP (r > 0.8). The nitrate-rich factor presents a lower correlation, as well as the sea/road salt one (0.3 < r < 0.6 for both OP's), whereas the secondary biogenic, primary biogenic, and sulfate-rich factors are slightly anti correlated with both OP's ( 0.6 < r < 0.3). Crustal dust correlation is not significant with respect to the AA test but presents low correlation to the DTT test (r = 0.15 and r = 0.35, respectively).

P13 l3, we added this paragraph:

Nevertheless, all these results contrast with those from simple univariate correlations between OP and sources. Indeed, the secondary biogenic source which is slightly anti-correlated to both OP's is in fact the second most redox-active source when considering intrinsic $OP_{DTT}$. On the contrary, the sulfate-rich factor is slightly anti-correlated to the OP AAv but present an intrinsic $OP_{AA}$ close to 0. The vehicular factor, which highly correlates with OP's is also the dominant source in terms of intrinsic OP's for both assays. Such results emphasize the real interest to go replace the simple univariate correlation by a more comprehensive statistical analysis when considering the contribution of the sources (or species) to the OP's.

5) A more thorough discussion should be made in the Intrinsic OP section, namely a comparison with other values found in the literature (even though the majority concerns PM2.5) and the use or not of an intercept in the linear regression model. Furthermore, it is stated that other studies also highlight the importance of the vehicular source to explain the OP. In Verma et al. (2015) even though HOA (representing traffic) correlates significantly with OP at some sites, the generated linear regression models do never include HOA, though in some cases the linear regression model include copper. It is known that copper may originate from brake wearing, but also it can be linked to other anthropogenic activities, such as industry and/or coal burning.

We agree that at some point a more in-depth discussion between our results and existing similar studies should be made. However, this was not the point of this paper which is focused on the methodology and only provides a case study for supporting it. An incoming study, applying this methodology to many other sites in France will be coming soon, and will compare our results to previous studies in much more details, and of course, Verma et al. (2015) will be discussed and compared in this paper.

Technical corrections: Title: "Oxydative" should be corrected to "Oxidative" Abstract, L1: ". . .induces cellular oxidative stress in vivo, leading to adverse. . ." P6, L19: ". . .DTTv shows larger values. . ." (delete "has") P7, L7: ". . .sources appear to be strongly correlated. . ." P14, L18: "...biomass burning and vehicular sources. . ."

We would like to thank again the reviewer for his/her careful review and corrections.

Please also note the supplement to this comment:
https://www.atmos-chem-phys-discuss.net/acp-2017-1053/acp-2017-1053-AC2-supplement.pdf